# Computational Design of Novel Griseofulvin Derivatives Demonstrating Potential Antibacterial Activity: Insights from Molecular Docking and Molecular Dynamics Simulation

**DOI:** 10.3390/ijms25021039

**Published:** 2024-01-15

**Authors:** Parisa Aris, Masoud Mohamadzadeh, Maaroof Zarei, Xuhua Xia

**Affiliations:** 1Department of Biology, University of Ottawa, 30 Marie Curie, P.O. Box 450, Ottawa, ON K1N 6N5, Canada; 2Department of Chemistry, Faculty of Sciences, University of Hormozgan, Bandar Abbas 71961, Iran; masoud.mohamadzadehh@gmail.com (M.M.); mzarei57@gmail.com (M.Z.); 3Nanoscience, Nanotechnology and Advanced Materials Research Center, University of Hormozgan, Bandar Abbas 71961, Iran; 4Ottawa Institute of Systems Biology, Ottawa, ON K1H 8M5, Canada

**Keywords:** antibacterial agent, docking, drug development, FtsZ, griseofulvin derivatives, molecular dynamics

## Abstract

In response to the urgent demand for innovative antibiotics, theoretical investigations have been employed to design novel analogs. Because griseofulvin is a potential antibacterial agent, we have designed novel derivatives of griseofulvin to enhance its antibacterial efficacy and to evaluate their interactions with bacterial targets using in silico analysis. The results of this study reveal that the newly designed derivatives displayed the most robust binding affinities towards PBP2, tyrosine phosphatase, and FtsZ proteins. Additionally, molecular dynamics (MD) simulations underscored the notable stability of these derivatives when engaged with the FtsZ protein, as evidenced by root mean square deviation (RMSD), root mean square fluctuation (RMSF), radius of gyration (Rg), and solvent-accessible surface area (SASA). Importantly, this observation aligns with expectations, considering that griseofulvin primarily targets microtubules in eukaryotic cells, and FtsZ functions as the prokaryotic counterpart to microtubules. These findings collectively suggest the promising potential of griseofulvin and its designed derivatives as effective antibacterial agents, particularly concerning their interaction with the FtsZ protein. This research contributes to the ongoing exploration of novel antibiotics and may serve as a foundation for future drug development efforts.

## 1. Introduction

A dramatic upsurge in bacterial infections coupled with a growing resistance to existing antibiotics poses a serious threat to the global healthcare system [1,2]. The majority of antibiotics were discovered during the golden age of antibiotic therapy, which started in 1928 with the discovery of penicillin and peaked in the middle of 1950 [3]. However, there has been a gradual decline in antibiotic discovery and development since then, leading to the emergence of drug resistance in many human infections and the current antimicrobial resistance crisis [2,4]. More than half of the known natural product antibiotic classes are produced by filamentous actinomycetes, with the remainder originating from other bacteria and fungi [5].

Griseofulvin is a fungal natural product and secondary polyketide metabolite that was initially isolated from *P. griseofulvum* in 1939 and has since been found in various other fungal species [6,7,8]. A comparative genome analysis of over 250 fungal species revealed that only seven out of the thirteen putative genes in the *gsf* cluster are conserved among most genomes and may play a crucial role in griseofulvin production [9]. Griseofulvin has been primarily employed to treat dermatophyte infections, but it has also garnered attention due to its potential in treating colon and breast cancer [10], inhibiting the replication of the hepatitis C virus (HCV) [11], and most recently, its potential therapeutic applications for COVID-19 [12].

Research has shown that griseofulvin can inhibit tubulin polymerization and microtubule dynamics, leading to the suppression of cell division and, ultimately, cell death [13,14]. To understand the molecular interactions of griseofulvin with various β-tubulin isotypes, molecular docking and molecular dynamics simulations were conducted recently [15]. Additionally, griseofulvin has been found to bind to keratin cytoskeleton intermediate filament proteins K8 and K18, altering keratin solubility and causing the formation of Mallory bodies (MBs) in hepatocytes [16]. While humans and rodents may experience hepatitis and MB formation following griseofulvin treatment, severe hepatitis is more common in rodents due to the stronger binding affinity of griseofulvin to K18 in rodents compared to humans [16].

Due to its low toxicity and wide-ranging applications, griseofulvin has garnered renewed interest in recent years. Previous studies have indicated its promising antibacterial activity against various bacteria, although its effects differ across bacterial species, and the underlying molecular mechanisms remain poorly understood [17]. Given the scarcity of newly discovered antibiotics in the past two decades, despite extensive screening of bacterial and fungal secondary metabolites, there is an urgent need to develop more effective antibiotics to address this global crisis.

Schiff bases, or imines, play a vital role in the synthesis of bioactive compounds, including β-lactams [18,19,20]. Key components like 6-aminopenicillanic acid (6-APA) and 7-aminocephalosporanic acid (7-ACA) are essential in the production of penicillins and cephalosporins [21]. Through chemical modifications of semi-synthetic 6-APA and 7-ACA, more than fifty commercial antibiotics have been developed and manufactured [22]. Previously reported, benzenesulfonyl hydrazones are known for their potential antibacterial activity, belonging to a class of compounds with diverse biological activities in medicine, primarily in the areas of antibacterial, antifungal, anticancer, and antidepressant effects [23]. Aromatic amine 3,4-dichloroaniline is exclusively used as an intermediate in the chemical industry for synthesizing the antibacterial agent triclocarban, particularly effective against Gram-positive bacteria, such as Staphylococcus aureus [24]. Reports suggest that the presence of a methylenedioxy ring in derivatives of methylenedioxy aniline may enhance their antibacterial activity [25]. Notably, morpholine, a strong alkali, is a crucial precursor in antibiotic production [26].

Theoretical studies play a crucial role in accelerating drug development and conserving resources. Several successful examples of computational studies in medicine showcase the transformative impact of theoretical studies on drug development and discovery. From identifying potential targets for the hepatitis C virus [27] to repurposing safe medicines for the treatment of multidrug-resistant tuberculosis bacteria [28], these advancements highlight the versatility of computational approaches. These successes underscore the potential of integrating computational tools into medical research, offering innovative solutions to some of the most challenging healthcare problems. Taking into account the antibacterial potential of the mentioned compounds, we aimed to design novel griseofulvin analogs as antibacterial agents and conducted in silico studies to investigate their binding affinities and stability towards potential bacterial targets. We hypothesized that griseofulvin and its newly designed derivatives might have the potential to interact favorably with the FtsZ protein in bacterial cells, considering its structural similarity to microtubules [29], which are the primary targets of griseofulvin.

## 2. Results and Discussion

In this research, we employed a pioneering molecular hybridization approach in the realm of drug design and development. This method involves the fusion of pharmacophoric elements extracted from various bioactive compounds, resulting in the production of novel hybrid compounds that exhibit elevated affinity and efficacy compared to the parent drugs [30]. In this study, the imino group, which consists of a nitrogen atom doubly bonded to a carbon atom, served as a versatile linker in molecular structures, providing versatility in synthetic strategies. Imino linkers are stable under a wide range of conditions, making them suitable for use in diverse chemical reactions without undergoing undesired side reactions. There have been previously reported instances of Schiff bases successfully synthesized through this reaction [31,32].

### 2.1. Structure Analysis of Novel Derivatives 

The MOL files of these compounds were used as input data for similarity searches, employing default parameter settings. Notably, none of the six designed compounds exhibited any similarity with existing pharmaceutical agents within the Clarivate Analytics Integrity database. Table 1 displays the two-dimensional (2D) structures of these newly designed compounds, along with calculated LogP values and ligand bond characteristics. It is worth mentioning that all the examined compounds had molecular weights falling within the range of 471.89 to 670.03 Daltons, with predicted LogP values ranging from 2.87 to 4.39. Each of the derivatives shared attributes such as possessing no more than 10 hydrogen bond acceptors (except for G2) and no more than five hydrogen bond donors. Even though derivative G2 does not fully comply with Lipinski’s rules, there are instances of commercially successful used drugs that exhibit non-compliance with these rules [33], suggesting a reconsideration of Lipinski’s rule of five.

To investigate the antibacterial activity of these designed derivatives and comprehend their potential mechanisms of action, we employed an in silico approach, involving molecular docking, molecular dynamics, and Way2Drugs Prediction of Activity Spectra for Substances (PASS). The prediction relies on analyzing the relationships between structure and activity for over 250,000 biologically active compounds. The overall accuracy of predictions is approximately 95% for the entire PASS training set [34]. In Table 2, the expected biological activities, particularly focusing on the antibacterial and phosphatase inhibitory effects of the griseofulvin derivatives, are presented. In this study, ciprofloxacin served as the reference drug for activity comparisons. Notably, the results reveal that derivative G2 exhibited antibacterial activity surpassing that of ciprofloxacin. Significantly, all the derivatives exhibited noteworthy potential as inhibitors of phosphatase enzymes, a subject that we will investigate more in-depth in this study. 

### 2.2. Molecular Docking Studies

Regarding this prediction, we investigate the potential of the derivatives concerning an important protein phosphatase found in bacteria that plays a pivotal role in regulating various cellular functions [35,36]. This tyrosine phosphatase (PTP) is an essential virulence factor in pathogenic bacteria such as *Staphylococcus aureus*, *Mycobacterium tuberculosis*, *Salmonella typhimurium*, and *Yersinia* sp. [36]. Notably, tyrosine phosphatase A (TbpA) from *Pseudomonas aeruginosa* serves as a positive regulator in the formation of biofilms by controlling the concentration of the second messenger cyclic diguanylic acid (c-di-GMP) [37]. In this study, derivatives G1–G6 have a strong affinity for tyrosine phosphatase, with binding energies ranging from −33.65 to −49.01 kcal mol^−1^. This observation aligns with the predictions made by Way2Drug.

FtsZ is identified as a bacterial homolog to eukaryotic tubulin, forming a dynamic structural framework for the Z-ring, strategically positioned at the midpoint of bacterial cells to govern the process of cytokinesis [29]. FtsZ has emerged as a primary target for antibacterial agents due to its indispensable role in cell division across a wide spectrum of bacteria while remaining non-essential in eukaryotes [38,39]. Despite its homology with eukaryotic tubulin, FtsZ shares only a modest 10–18% sequence similarity with tubulin, making it less toxic to eukaryotic cells [40]. Our docking analysis at the inter-domain cleft (IDC) has revealed remarkably strong binding energies between griseofulvin and its derivatives with FtsZ, ranging from −29.48 to −38.59 kcal mol^−1^.

MurA represents a pivotal target for bacterial replication inhibition since it catalyzes the initial step in peptidoglycan synthesis. In this study, it was observed that the 3-aminopropyl morpholine derivative displayed the most robust binding affinity with MurA, whereas ciprofloxacin did not exhibit any interaction or affinity towards this particular target. Ciprofloxacin, which belongs to the second generation of fluoroquinolones, is known for inhibiting bacterial DNA gyrase and topoisomerase IV [41].

Ciprofloxacin displayed a commendable binding affinity of −38.22 kcal mol^−1^ with GyrB, while G6 exhibited the most favorable binding energy of −41.55 kcal mol^−1^ for this target. Of particular interest, two novel derivatives G3 and G6 demonstrated elevated affinities for DNA topoIV, with binding energies of −40.75 and −40.85 kcal mol^−1^, surpassing the binding affinity of ciprofloxacin, which was lower when compared to the designed derivatives.

To initiate DNA replication, cellular organisms rely on specialized RNA polymerases called primases to synthesize RNA primers [42]. In this study, it was found that G5 and G2 derivatives displayed the most favorable affinity with binding energies of −34.95 and −28.16 kcal mol^−1^ in their interaction with primase, whereas both griseofulvin and ciprofloxacin showed the weakest binding energies, measuring at −17.54 and −18.42 kcal mol^−1^, in contrast to the new derivatives.

Penicillin-binding proteins (PBPs) are essential proteins in the cell wall peptidoglycan synthesis. Consequently, targeting these proteins could represent one of the most effective strategies for disrupting cell wall biosynthesis. Notably, all of the newly designed derivatives exhibited the highest affinity for PBP2. In the molecular docking analysis, it was evident that G2 and G3 displayed particularly strong binding affinities, with binding energies of −55.29 and −57.04 kcal mol^−1^, surpassing the binding energies of other derivatives, which ranged from −40.82 to −52.17 kcal mol^−1^ for PBP2. A comprehensive summary of the binding affinities of these novel analogs, in comparison to the control compounds, for the selected targets is provided in Table 3. Ciprofloxacin served as a control in this study.

The analysis of hydrogen bonds and hydrophobic interactions involving griseofulvin derivatives, which were subjected to docking with the antibacterial targets listed in Table 4, revealed interesting insights. The outcomes of molecular docking indicated that hydrophobic interactions, including Alkyl, Pi-Anion, and van der Waals forces, played dominant roles in most targets, such as murA, DNA topoIV, and GyrB. In contrast, hydrogen bonds emerged as significant contributors in interactions with tyrosine phosphatase, FtsZ, and PBP2. Within PBP2, for instance, the residues Thr216, Arg241, and Ser249 established hydrogen bonds with the carbonyl group of the ligands. In the case of tyrosine phosphatase, the amino acid Arg106 appeared to be notably vital in facilitating hydrogen bond interactions with the ligands. Furthermore, the amino acid Pro248 in the ftsz protein was observed to form carbon–hydrogen bonds with derivatives G1–G3 and griseofulvin.

### 2.3. MD Simulations

The top three docked complexes were subjected to 100 ns of MD simulations to investigate the protein structural stability and conformational changes. Noteworthy, all derivatives exhibited the strongest binding affinity with PBP2. The molecular docking analysis of tyrosine phosphatase is in good agreement with the biological activity prediction by PASS. Furthermore, the derivatives displayed favorable binding to ftz, a microtubule homolog that griseofulvin might target. The simulation trajectory files were examined for root mean square deviation (RMSD), root mean square fluctuation (RMSF), radius of gyration (Rg), and solvent-accessible surface area (SASA).

#### 2.3.1. Structural Dynamics of PBP2 (1VQQ)

Molecular dynamics simulations were used to study the complex of the PBP2 enzyme and the novel derivatives. The root mean square deviation (RMSD) was calculated to evaluate the stability of these complexes. The RMSD of the enzyme in the absence of ligand exhibited fluctuations around an average value of 0.35 nm. Upon binding of analogs, the RMSD values increased from 0.36 to 0.91 nm, indicating conformational changes in the enzyme upon complex formation. Smaller RMSD values correspond to more steadfast protein structures. RMSF (root mean square fluctuation) plots were analyzed for both the ligand-free and ligand-bound states. In the absence of the ligand, the enzyme exhibited an average RMSF of 0.19 nm. However, upon ligand binding, the average RMSD values of G2, G4, and G6 were decreased to 0.18, 0.14, and 0.13 nm, respectively, indicating the stability of complexes during the simulation. The radius of gyration (Rg) provides insight into the protein’s structural compactness. When Rg is smaller, it suggests a more tightly packed protein structure. In the case of the complexes, the average Rg values remained mostly unchanged or slightly decreased (Figure 1), indicating the ligand binding did not significantly alter the structural compactness of PBP2. Solvent-accessible surface area analysis (SASA) was used to measure the surface area of a compound that is accessible to solvent molecules. The mean SASA measurements for derivatives remained almost unchanged when they bound to PBP2. This observation implies that the protein has undergone conformational changes at a comparatively slow rate and no equilibrium has been reached, as evidenced by its solvent accessibility. Overall, the analysis suggests that no equilibrium has been reached for this target during the last 20 ns of MD simulations (Table 5). 

#### 2.3.2. Structural Dynamics of Tyrosine Phosphatase (2M3V)

The MD analysis showed the average RMSD values for all derivatives but G3 decreased up to 0.15 nm when binding to tyrosine phosphatase (Table 6). This finding illustrates the stability of these complexes within the active pocket of tyrosine phosphatase, with only a slight increase in residual fluctuations (Figure 2B). Ciprofloxacin demonstrated a significantly elevated root mean square deviation (RMSD) from 0.73 to 0.85 nm, which is in line with expectations, as tyrosine phosphatase is not a conventional molecular target for ciprofloxacin. The average RMSF and Rg values of the complexes exhibited negligible alterations, indicating their structural packing closely resembles that of tyrosine phosphatase alone. The average SASA values for the derivatives exhibited a notable increase, going from 88 to 118 nm². This observation indicates that the internal residues within tyrosine phosphatase become more accessible to solvents and that there is an expansion in protein volume upon the binding of the derivatives.

#### 2.3.3. Structural Dynamics of FtsZ(3VOB)

The RMSD trajectory assesses the deviation of complexes from the reference structure. When ligands bind to the receptor-binding pocket, they contribute to the conformational stability of the macromolecular system. As shown in Table 7, all complexes displayed a lower RMSD compared to FtsZ alone, signifying the structural stability of the designed derivatives within the active pocket of FtsZ protein. The average RMSF values for the complexes in the final 20 ns of the MD simulations indicated that the fluctuations in their constituent residues remained below 0.2 nm, which is considered satisfactory. The Rg plot indicates that there are no notable variations in the arrangement of FtsZ when ligands are bound. From the SASA plot in Figure 3D, it is evident that the complexes exhibit higher SASA values compared to the standalone FtsZ protein. This indicates an expansion in the solvent-accessible surface area, contributing to enhanced protein stability. 

## 3. Conclusions

In this study, a series of novel griseofulvin derivatives were designed and evaluated for antibacterial activity using computer-aided approaches. Data analysis of compounds through a similarity search in the Clarivate Analytics Integrity portal validated their novelty regarding antibacterial activity. Structure analysis of derivatives revealed the presence of substituents like β-lactam moiety (G1 and G2), hydrazine (G3), sulfonyl group (G4), chlorine (G5), and methylenedioxy ring (G6) groups are beneficial for antibacterial activity. Molecular docking studies and predictions using the PASS server indicated that the derivatives exhibited strong binding modes to the PBP2, tyrosine phosphatase, and FtsZ proteins. However, the MD analysis demonstrated that these complexes lack stability upon binding and do not induce any significant changes in the protein’s conformation of PBP2 throughout the simulation. Noteworthily, derivatives G3 and G5 displayed the greatest stability within the active site of tyrosine phosphatase, as evidenced by a decrease in RMSD and an increase in SASA, suggesting enzyme conformational changes upon complex formation. MD simulations unveiled the remarkable stability of the studied derivative when interacting with FtsZ, as indicated by RMSD, RMSF, Rg, and SASA values. This observation aligns with expectations since griseofulvin targets microtubules in eukaryotes, and FtsZ serves as a microtubule homolog in prokaryotes. These findings underscore the potential efficacy of griseofulvin and its derivatives against bacterial targets, especially the FtsZ protein. This study is constrained by the small number of designed compounds. Notwithstanding this limitation, this research has the potential to offer insights into the advancement of novel derivatives and the enhancement of the antibacterial activity of griseofulvin through a hybridization approach in derivative development. Furthermore, this investigation has the potential to set a benchmark for future endeavors in the design of griseofulvin inhibitor compounds targeting bacterial proteins, laying the foundation for the development of promising antibacterial agents.

## 4. Materials and Methods

### 4.1. Design Novel Derivatives

In this study, we employed molecular hybridization techniques for drug design and development [30]. This method entails the fusion of pharmacophoric elements from different bioactive compounds to generate novel hybrid compounds with superior affinity and effectiveness compared to their parent drugs. Clarivate Analytics Integrity drug discovery and development portal, available at https://integrity.clarivate.com/ (accessed on 16 November 2023), was used to assess the novelty of designed compounds.

### 4.2. Ligand and Receptor Preparation

We initiated the process by designing the two-dimensional structures of ligands using ChemDraw Software https://revvitysignals.com/products/research/chemdraw, accessed on 16 November 2023. Subsequently, we created and optimized the three-dimensional molecular structures of these compounds with Chem3D. Ligand preparation was conducted using the Ligprep panel in Maestro. For the receptors, we obtained the three-dimensional protein structures in .pdb format from the RCSB Protein Data Bank (PDB). The PDB IDs corresponding to the tyrosine phosphatase, filamenting temperature-sensitive mutant Z (FtsZ), UDP-N-acetylglucosamine enolpyruvyl transferase (murA), penicillin-binding protein 2 (PBP2), DNA gyrase subunit B (GyrB), DNA topoIV, and primase proteins were 2M3V, 3VOB, 1UAE, 1VQQ, 4PRV, 1S16, and 1DDE, respectively. These proteins are widely recognized as prominent drug targets. All protein structures were subjected to preparation using the Protein Preparation wizard in Maestro.

### 4.3. Drug-Like Properties of the Ligands

To determine the drug likeness of the molecules, we employed Lipinski’s rule of five parameters, which include criteria such as a molecular weight less than 500 Da, no more than 5 hydrogen bond donors, fewer than 10 hydrogen bond acceptors, and a LogP value not exceeding 5 [43]. Lipinski’s rule of five parameters was obtained using the SwissADME web-based program (www.swissadme.ch/index.php, accessed on 16 November 2023) [44]. 

### 4.4. Biological Activity Predictions

The biological activity of the designed derivatives was predicted using the PASS Online web service in the Way2Drug web portal [45]. The predictions are grounded in structure–activity relationships (SAR) derived from a diverse training set encompassing various chemical classes. The PASS training set includes approximately 7000 antibacterial drugs, and the prediction accuracy for this activity is approximately 0.92. We provided the structural formulas of the compounds in MOL structure data files as input to PASS, which yielded predictions, including probabilities for the presence of activity (Pa).

### 4.5. In Silico Molecular Docking

The molecular docking analyses were carried out using the Schrödinger Suite 2020-3. To determine the binding sites on the targeted proteins, we utilized the computed Atlas of Surface Topography of Proteins (CASTp) server and constructed a grid box around the identified binding pocket, maintaining the default size of 20 Å. Molecular mechanics generalized born surface area (MM-GBSA) method was used to calculate binding free energy in this study [46]. The interactions between the ligands and receptors were visualized using Discovery Studio Visualizer v.20.

### 4.6. Molecular Dynamic Simulations

To validate the docking results, molecular dynamics simulations (MD) were performed. The GROMACS-2021 software package was used to analyze the single protein and the two best protein–ligand complexes through 100 ns simulations [47] with an AMBER99SB force field [48]. The force field parameters for the ligands were generated by the Ante Chamber PYthon Parser interface (ACPYPE) [49]. The solvation of complexes employed the TIP3P water model, with the addition of sodium and chloride ions to maintain charge neutrality. Periodic boundary conditions (PBC) were applied, and an energy minimization was performed at 1000 kJ/mol/nm. The systems were equilibrated in the NVT and NPT ensembles for 1 ns, with Na+ and Cl− counter-ions added to achieve charge neutrality and a 0.15 M ionic strength. The temperature and pressure were controlled at 310 K and 1.0 bar, respectively, using the Nose–Hoover thermostat [50], and the Berendsen barostat [51]. Trajectories were saved at 2 ps intervals and analyzed post-simulation, which included parameters such as root mean square deviation (RMSD), root mean square fluctuations (RMSF), radius of gyration (Rg), and solvent-accessible surface area (SASA). 

## Figures and Tables

**Figure 1 ijms-25-01039-f001:**
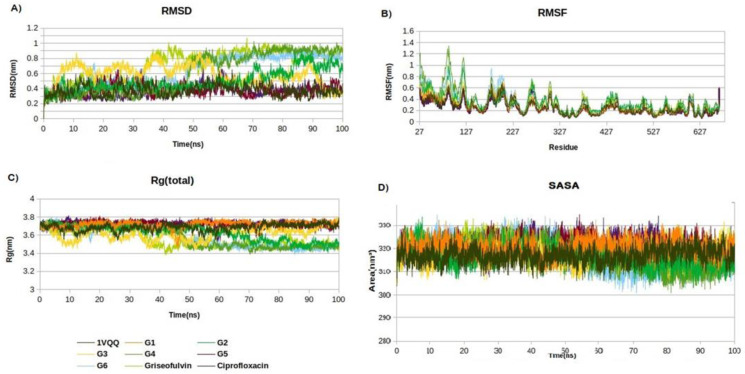
Structural dynamics of PBP2 protein. (**A**) Root mean square deviation (RMSD), (**B**) root mean square fluctuations (RMSF), (**C**) radius of gyration (Rg) plot, and (**D**) solvent-accessible surface area (SASA).

**Figure 2 ijms-25-01039-f002:**
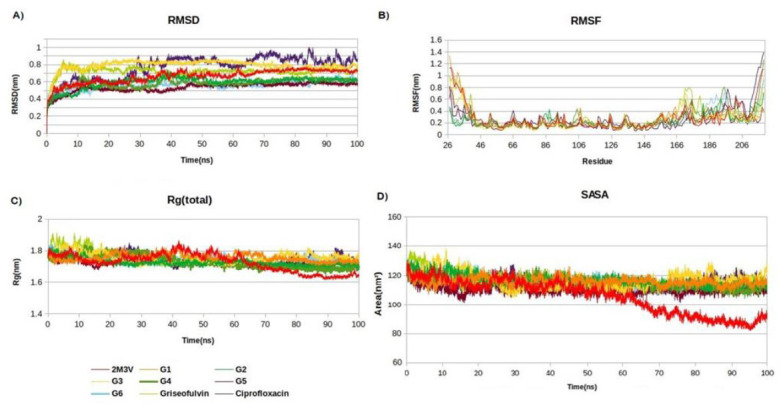
Structural dynamics of tyrosine phosphatase protein. (**A**) Root mean square deviation (RMSD), (**B**) root mean square fluctuations (RMSF), (**C**) radius of gyration (Rg) plot, and (**D**) solvent-accessible surface area (SASA).

**Figure 3 ijms-25-01039-f003:**
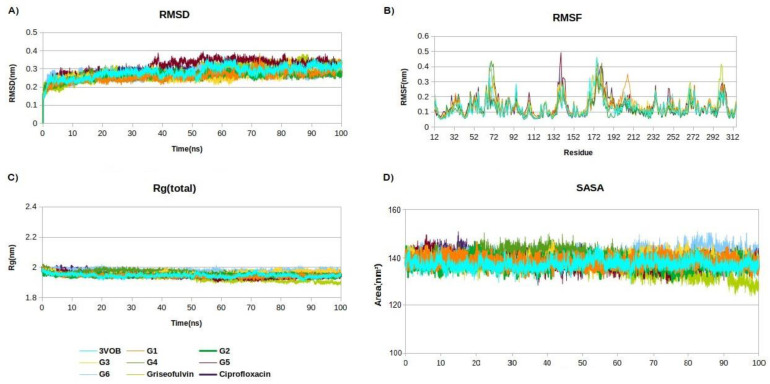
Structural dynamics of ftsZ protein. (**A**) Root mean square deviation (RMSD), (**B**) root mean square fluctuations (RMSF), (**C**) radius of gyration (Rg) plot, and (**D**) solvent-accessible surface area (SASA).

**Table 1 ijms-25-01039-t001:** Chemical 2D structures of the novel derivatives along with the Lipinski analysis.

	RNH_2_	Lipinski Analysis	2D Structure
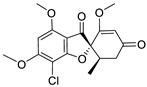 +RNH_2_Griseofulvin	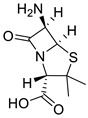 6-APA	Molecular weightLipophilicity H bond donorsH bond acceptorsViolations	551.013.191101	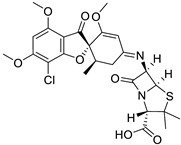 (G1)
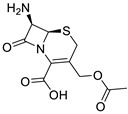 7-ACA	Molecular weightLipophilicity H bond donorsH bond acceptorsViolations	607.032.871122	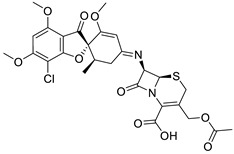 (G2)
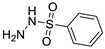 Benzenesulfonyl hydrazine	Molecular weightLipophilicity H bond donorsH bond acceptorsViolations	506.963.25181	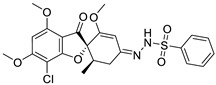 (G3)
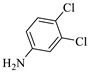 3,4-Dichloroaniline	Molecular weightLipophilicity H bond donorsH bond acceptorsViolations	496.774.19080	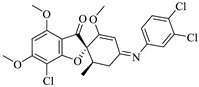 (G4)
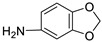 3,4-Methylenedioxy aniline	Molecular weightLipophilicity H bond donorsH bond acceptorsViolations	471.894.19080	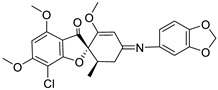 (G5)
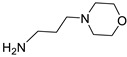 3-Aminopropyl morpholine	Molecular weightLipophilicity H bond donorsH bond acceptorsViolations	478.974.39080	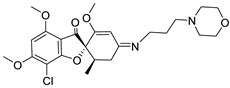 (G6)

**Table 2 ijms-25-01039-t002:** Prediction of antibacterial and phosphatase inhibitor activities (Pa) for griseofulvin and derivatives G1–G6.

Compounds	Antibacterial Activity (Pa)	Phosphatase Inhibitor Activity (Pa)
G1	0.435	0.803
G2	0.718	0.712
G3	<0.1	0.754
G4	0.304	0.755
G5	0.259	0.727
G6	0.254	0.735
Griseofulvin	0.342	0.803
Ciprofloxacin	0.588	0.301

**Table 3 ijms-25-01039-t003:** The binding energies (kcal mol^−1^) of griseofulvin analogs towards bacterial targets from *Pseudomonas aeruginosa*, *Staphylococcus aureus*, and *Escherichia coli*.

	Tyrosine Phosphatase (2M3V)	FtsZ (3VOB)	murA (1UAE)	PBP2 (1VQQ)	GyrB (4PRV)	DNA topoIV (1S16)	Primase (1DDE)
G1	−33.65	−29.48	−24.05	−42.09	−30.73	−14.22	−20.02
G2	−33.81	−34.90	−36.90	−55.29	−35.83	−37.57	−28.16
G3	−49.01	−35.62	−34.79	−57.04	−27.77	−40.75	−18.42
G4	−42.85	−35.62	−41.32	−52.17	−38.09	−27.22	−26.86
G5	−36.25	−37.85	−35.57	−49.87	−27.32	−20.58	−34.96
G6	−45.65	−38.59	−46.48	−46.17	−41.55	−40.85	−24.02
Griseofulvin	−22.37	−38.09	−37.94	−40.82	−30.88	−18.85	−17.54
Ciprofloxacin	−25.98	−31.46	0	−43.65	−38.22	−26.65	−18.42

**Table 4 ijms-25-01039-t004:** Molecular docking details of the novel derivatives with antibacterial targets.

Targets	Ligands	Binding Energy (kcal/mol)	H Bonds	Hydrophobic Interaction
Tyrosine phosphatase (2M3V)	G1	−33.65	Arg106	His103, Arg106, Arg117, Leu204
	G2	−33.81	Arg106, Arg117, Ser200	Ser99, Pro101, His103
	G3	−49.01	Arg106, Glu111	Phe81, Ile82, Pro101, His103, Arg106
	G4	−42.85	Arg106	Val79, Ser80, Phe81, Ile82, Ser99, Pro101, His103
	G5	−36.25	Arg106, Ser200	Pro101, Glu111, Ala191, Ala203, Leu204
	G6	−45.65	Arg106	Ser80, Phe81, Ile82, Lys83, Ser99, Glu111, Gly195, Phe202
	Griseofulvin	−22.37	Arg106	Arg106, Glu111, Ala191, Ser198
	Ciprofloxacin	−25.98	Leu100, Arg106	Leu100, Pro101, Glu111, Gly195
ftsZ (3VOB)	G1	−29.48	Val230, Lys243, Leu249	Ile228, Pro248
	G2	−34.90	Gln192	Glu185,Gly227, Ile228, Ala244, Ser246, Pro248, Leu250,
	G3	−35.62	-	Val189, Ile228, Val230, Pro248
	G4	−35.62	Gln192	Lys184, Glu185, Ser246, Leu225, Ala244, Leu250
	G5	−37.85	Ser246	Glu185, Asn188, Gly227, Lys243
	G6	−38.59	-	Met226, Gly227, Val230, Lys243, Ser246
	Griseofulvin	−38.09	Asn188, Gln192, Leu249	Val230, Lys243, Pro248
	Ciprofloxacin	−31.46	-	Gly227, Ile228, Lys243, Ser246, Pro248, Leu249
murA (1UAE)	G1	−24.05	Asn23, Cys115	Lys22, Arg120, Gly114, Pro121, Val327, Phe328, Arg397
	G2	−36.90	Arg120, Gly164	Leu26, Arg91, Ala92, Ile94, Trp95, His125, Val163, Arg120, Lys160, Phe328
	G3	−34.79	Arg120, Lys160	Asn23, Arg91, Alys160, Val161, Val163, Asp305, Phe328, Glu329,
	G4	−41.32	Ser162, Val163, Gly164	Arg91, Trp95, Arg120, Lys160, Val 161, Val 327, Phe328
	G5	−35.57	Thr326	Arg91, Tpr95, Arg120, His125, Val161, Ser162, Val163, Ala297, Pro298, Phe328
	G6	−46.48	Cys115, Val163, Arg120	Asp305, Val327, Phe328
	Griseofulvin	−37.94	Lys22, Arg120	Val163, Pro298, Val327, Phe328
	Ciprofloxacin	0	-	-
PBP2 (1VQQ)	G1	−42.09	Thr216, Ser240, Arg241	Tyr196, Ile192, Lys215
	G2	−55.29	Thr216, Arg241	Ile192, Tyr196, Lys215, Val277
	G3	−57.04	Thr216, Ser240, Arg241	Lys148, Ser149, Arg241, His293
	G4	−52.17	Thr165, Thr216	Arg151, Lys215, Ser240, Pro258, Tyr373
	G5	−49.87	Thr216, Ser240, Arg241	Lys148, Ser149, Thr165, Thr216, Arg241, Val277
	G6	−46.17	Ser240, Arg241	Lys148, Ser149, Thr165, Thr216, Arg241, His293, Met372
	Griseofulvin	−40.82	Ser240, Arg241	Arg241, Val277, His293
	Ciprofloxacin	−43.65	Arg151, Glu170, Thr216, Thr238, Ser240	Val277
GyrB (4PRV)	G1	−30.73	His 38	Thr34, His38, Ile27, Tyr267, Pro274, Arg276, Asp338,
	G2	−35.83	Cys268, Arg276	Phe41, Tyr267, Cys268, Phe269, Pro274, Arg276
	G3	−27.77	His38, Arg276, Asp338	Lys189, Arg190, Glu193, Pro274, Tyr267, Asp338
	G4	−38.09	-	His38, Phe41, Tyr267, Pro274, Arg276
	G5	−27.32	-	Lys189, Glu193, Tyr267, Pro274, Arg276
	G6	−41.55	-	His38, Phe41, Lys189, Tyr228, Pro274, Arg276, Asp338
	Griseofulvin	−30.88	Cys268	Ile266, Tyr267, Cys268, Phe269, Pro274, Arg276
	Ciprofloxacin	−38.22	His38, Glu193	Lys189, Pro274, Arg276
DNA topoIV (1S16)	G1	−14.22	Arg1031, Arg1183	Arg1031, Arg1183, Thr1264, Ser1266, Met1274, Tyr1318
	G2	−37.57	Thr1264, Gln1037	Arg1031, Arg1183, His1186, Leu1262, Thr1264, Ser1266, Thr1273
	G3	−40.75	Ser1182, Thr1264	Met1274
	G4	−27.22	-	Ser1182, Arg1183, His1186, Val1187, Met1274
	G5	−20.58	Arg1183	His1034, Arg1183, His1186, Val1187, Met1274
	G6	−40.85	Lys1189	His1034, Arg1183, His1186, Val1187, Tyr1222, Met1274
	Griseofulvin	−18.85	Arg1183	His1034, Arg1183, His1186, Val1187, Met1274, Pro1272
	Ciprofloxacin	−26.65	Asp1028	Arg1183, His1186, Val1187, Pro1272, Met1274
Primase (1DDE)	G1	−20.02	Lys229, Asp345	Arg146,Arg221, Lys229, Arg221,Tyr230, Asn232, Gly266, Tyr267, Met268, Leu285, Thr287, Asp309, Asp345
	G2	−28.16	Arg146, Lys229	Arg146, Arg221, Lys229, Tyr230, Asn232, Gly266, Tyr267, Met268, Asp269, Gly286, Thr287, Asp309
	G3	−18.42	Asp345	Tyr230, Gly266, Tyr267, Met268, Asp269, Ser284, Asp309, Asp345
	G4	−26.86	Lys229	Tyr230, Glu265, Tyr267
	G5	−34.96	Lys229	Lys229, Tyr230, Asn232, Glu265, Gly266, Tyr267, Met268, Leu285
	G6	−24.02	Arg221, Lys229, Tyr230	Tyr142, Arg146, Gly266, Met268
	Griseofulvin	−17.54	-	Lys229, Tyr230, Gly266, Met268, Asp345, Asp347
	Ciprofloxacin	−18.42	Tyr230, Tyr267, Asp345	Tyr230, Asn232, Asp269, Gly286, Asp345

**Table 5 ijms-25-01039-t005:** The average of RMSD, RMSF, Rg, and SASA for PBP2 during the last 20 ns of MD simulations.

Complex	Mean RMSD (nm)	Mean RMSF (nm)	Mean Rg (nm)	Mean SASA (nm²)
PBP2	0.35 ± 0.05	0.19 ± 0.1	3.7 ± 0.0	319 ± 3.3
G1	0.36 ± 0.05	0.21 ± 0.1	3.7 ± 0.0	320 ± 3.2
G2	0.71 ± 0.08	0.18 ± 0.08	3.5 ± 0.0	313 ± 3
G3	0.49 ± 0.12	0.25 ± 0.1	3.6 ± 0.0	319 ± 3.1
G4	0.91 ± 0.03	0.14 ± 0.05	3.4 ± 0.0	310 ± 3
G5	0.38 ± 0.06	0.21 ± 0.1	3.7 ± 0.0	320 ± 3.3
G6	0.84 ± 0.03	0.13 ± 0.05	3.4 ± 0.0	312 ± 3
Griseofulvin	0.84 ± 0.06	0.17 ± 0.07	3.5 ± 0.0	318 ± 4
Ciprofloxacin	0.39 ± 0.04	0.19 ± 0.08	3.7 ± 0.0	317 ± 3

**Table 6 ijms-25-01039-t006:** The average of RMSD, RMSF, Rg, and SASA for tyrosine phosphatase protein during the last 20 ns of MD simulations.

Complex	Mean RMSD (nm)	Mean RMSF (nm)	Mean Rg (nm)	Mean SASA (nm²)
Tyrosine phosphatase	0.73 ± 0.01	0.12 ± 0.7	1.6 ± 0.1	88 ± 2
G1	0.66 ± 0.01	0.15 ± 0.09	1.7 ± 0.1	114 ± 2
G2	0.62 ± 0.01	0.13 ± 0.06	1.7 ± 0.1	113 ± 3
G3	0.77 ± 0.01	0.18 ± 0.1	1.6 ± 0.1	118 ± 2
G4	0.62 ± 0.01	0.11 ± 0.05	1.7 ± 0.1	110 ± 2
G5	0.58 ± 0.01	0.13 ± 0.07	1.7 ± 0.1	109 ± 2
G6	0.61 ± 0.03	0.16 ± 0.13	1.7 ± 0.1	113 ± 2
Griseofulvin	0.72 ± 0.02	0.15 ± 0.12	1.7 ± 0.1	114 ± 2
Ciprofloxacin	0.85 ± 0.04	0.20 ± 0.17	1.7 ± 0.1	117 ± 2

**Table 7 ijms-25-01039-t007:** The average of RMSD, RMSF, Rg, and SASA for FtsZ protein during the last 20 ns of MD simulations.

Complex	Mean RMSD (nm)	Mean RMSF (nm)	Mean Rg (nm)	Mean SASA (nm²)
FtsZ	0.31 ± 0.01	0.11 ± 0.04	1.9 ± 0.0	136 ± 2
G1	0.28 ± 0.01	0.12 ± 0.05	1.9 ± 0.01	137 ± 2
G2	0.27 ± 0.01	0.09 ± 0.04	1.9 ± 0.0	137 ± 3
G3	0.29 ± 0.01	0.11 ± 0.05	1.9 ± 0.0	138 ± 2
G4	0.27 ± 0.01	0.10 ± 0.05	1.9 ± 0.0	138 ± 2
G5	0.33 ± 0.01	0.10 ± 0.04	1.9 ± 0.0	136 ± 2
G6	0.28 ± 0.01	0.12 ± 0.06	1.9 ± 0.0	143 ± 2
Griseofulvin	0.33 ± 0.01	0.10 ± 0.04	1.9 ± 0.0	130 ± 3
Ciprofloxacin	0.29 ± 0.01	0.11 ± 0.04	1.9 ± 0.01	138 ± 2

## Data Availability

Data are contained within the article.

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
