# Peer review of "Computational Design of Novel Griseofulvin Derivatives Demonstrating Potential Antibacterial Activity: Insights from Molecular Docking and Molecular Dynamics Simulation"

_ijms, 2024, doi:10.3390/ijms25021039_

Round 1

Reviewer 1 Report

Comments and Suggestions for Authors

The manuscript entitled "In silico Design Novel Griseofulvin Derivatives with Potential Antibacterial Activity based on Molecular Docking and Molecular Dynamics Simulation Study" (Manuscript ID ijms-2751227) presents a computational study of 6 griseofulvin derivatives, with the aim of identifying novel antibiotics that can serve as a basis for future drug development efforts. The authors design the new structures, predict their antibacterial activity, as well as their binding affinity and stability against 6 protein targets.

The methodology used in this research comprises, the design of chemical structures, as well as calculations of the parameters included in the Lipinski rules, with the intention of ensuring their druggability. On the other hand,  molecular modeling techniques such as molecular docking and molecular dynamics simulations are used to identify the intermolecular interactions and the stability of the protein-ligand complexes.

The review of the manuscript gives rise to the following comments and suggestions:

The design approach chosen to propose new structures (molecular hybridization) cannot be considered innovative, but a drug design technique used for 20 years that, by combining pharmacophore residues, has resulted in new dual drugs. This fact is not consistent with the research approach nor with the final conclusions. 

The conclusions are severely limited by the small number of compounds with which the study was conducted. In addition, two of the six compounds designed (G1 and G2) do not meet all of Lipinski's rules, which further reduces the contributions to the proposed research.

Tables of molecular docking model results with so many intermolecular interaction data are not necessary, as the molecular dynamics simulations below only allow highlighting the stability of the ftsZ protein. 

As for the antibiotic activity tables, it is advisable to include antibiotic drugs as activity references, so that the relevance of the results is evident.

The computer programs used in this study (such as GROMACS, AMBER, ...), should be cited in the biobliography.

Finally, in studies with computational design tools it is advisable that they are supported by some experimental model. It is advisable to use some bioassay of the proposed compounds, or at least some bibliographic reference with these data to support it.

Author Response

The manuscript entitled "In silico Design Novel Griseofulvin Derivatives with Potential Antibacterial Activity based on Molecular Docking and Molecular Dynamics Simulation Study" (Manuscript ID ijms-2751227) presents a computational study of 6 griseofulvin derivatives, with the aim of identifying novel antibiotics that can serve as a basis for future drug development efforts. The authors design the new structures, predict their antibacterial activity, as well as their binding affinity and stability against 6 protein targets.

The methodology used in this research comprises, the design of chemical structures, as well as calculations of the parameters included in the Lipinski rules, with the intention of ensuring their druggability. On the other hand, molecular modeling techniques such as molecular docking and molecular dynamics simulations are used to identify the intermolecular interactions and the stability of the protein-ligand complexes.

The review of the manuscript gives rise to the following comments and suggestions:

The design approach chosen to propose new structures (molecular hybridization) cannot be considered innovative, but a drug design technique used for 20 years that, by combining pharmacophore residues, has resulted in new dual drugs. This fact is not consistent with the research approach nor with the final conclusions.

Yes, you are right but we did not mean that we innovated this method as you said it is being used for many years in drug design field as it was cited in line 104. The word “Innovative” was removed from the line 93 to avoid any confusion. Moreover, the lines 363-371 are reworded in the conclusion.

The conclusions are severely limited by the small number of compounds with which the study was conducted. In addition, two of the six compounds designed (G1 and G2) do not meet all of Lipinski's rules, which further reduces the contributions to the proposed research.

We agree. The limitation of limited studies compounds is added to the lines 362-371. Moreover, as mentioned in lines 175-178, even though derivative G2 does not fully comply with Lipinski's rules, there are instances of commercially successful used drugs that exhibit non-compliance with these rules, suggesting a reconsideration of Lipinski’s Rule of Five.

Tables of molecular docking model results with so many intermolecular interaction data are not necessary, as the molecular dynamics simulations below only allow highlighting the stability of the ftsZ protein.

We incorporated the specifics of the molecular docking analysis, revealing promising binding affinities and favorable interactions of the derivatives with the investigated antibacterial targets. However, molecular dynamics illustrated the stability of ftsz protein over time. In summary, molecular docking is valuable for predicting binding modes and affinities, while molecular dynamics simulations provide a more detailed understanding of the dynamic behavior of molecular complexes. Together, these techniques offer a comprehensive view of molecular interactions and are often used to gain a more accurate and insightful depiction of the studied systems

As for the antibiotic activity tables, it is advisable to include antibiotic drugs as activity references, so that the relevance of the results is evident.

The activity of ciprofloxacin has been included in Table 2 as a reference drug. The results reveal that derivative G2 exhibited antibacterial activity surpassing that of ciprofloxacin. This statement was added to the lines 191-193.

The computer programs used in this study (such as GROMACS, AMBER, ...), should be cited in the biobliography.

The Amber reference was added to the line 148. Gromacs and other computer programs were added to the 2.6. Molecular Dynamic Simulations section.

Finally, in studies with computational design tools it is advisable that they are supported by some experimental model. It is advisable to use some bioassay of the proposed compounds, or at least some bibliographic reference with these data to support it.

Although this special issue "Advances in Molecular Modeling, Docking and Simulations of Protein Structure" emphasizes the in silico analysis, the reviewer's suggestion on experimental evidence is highly relevant and well appreciated. In our next project, we plan to perform experimental studies on the designed griseofulvin derivatives based on this study.

Reviewer 2 Report

Comments and Suggestions for Authors

The manuscript has been well written and constructed so i agree to accept the manuscrip

Author Response

Thanks

Reviewer 3 Report

Comments and Suggestions for Authors

In this article, the authors have discussed in silico design novel griseofulvin derivatives with potential antibacterial activity based on molecular docking and molecular dynamics simulation study. The authors have made decent efforts to come up with a research article using in silico tools. In the reviewer’s opinion, the manuscript has been organized appropriately. The reviewer feels that a potential reader can easily understand the given details by consulting the main data outlined in the current version of the manuscript with few exceptions. The write-up is clear and to the point which is crucial for an article with few exceptions. Although there are several published articles closely related to this piece of study in general, the current version of the manuscript can bring a novelty in the existing literature however there are some laps that should be addressed before publication of this study. There are some suggestions related to the manuscript and authors should take into account these suggestions before resubmission.

  1. In its current state, the title itself should be rephrased and needs careful attention.
  2. The author revealed that their findings collectively suggest the promising potential of griseofulvin and its designed derivatives as effective antibacterial agents, particularly concerning their interaction with the ftsZ protein. Similarly, Table 2 shows the prediction of antibacterial and phosphatase inhibitor activities (Pa) for griseofulvin and derivatives G1-G6. The findings of this study have not been biologically validated not even a single type of testing ws performed. I would gladly reconsider revision if the author can provide biological validation of the claimed findings.
  3. The introduction section lacks objectivity and should be revised with more clarity. While it is undeniable that theoretical studies play a critical role in accelerating drug development and conserving resources, the section could benefit from providing specific examples of such studies. For instance, it could mention instances where extensive theoretical studies have led to the qualification of biological molecules from hit to lead. This would add more weight to the argument and make the section more objective.
  4. The authors hypothesized the notion that griseofulvin and its newly devised analogs could potentially interact favorably with the ftsZ protein in bacterial cells, given their structural resemblance to microtubules. Nevertheless, no empirical evidence/ biological testing was presented to verify this hypothesis in the current study.
  5. In the opinion of the reviewer, a limited number of theoretical analogs were designed by the author, six in total. This low number is insufficient to generate a meaningful SAR for the generalization of activity. The authors should discuss the limitations of being limited to such small theoretical analogs.
  6. A few types of amines were chosen for the design of G3-G6. What was the crucial factor in selecting these amines??

Author Response

In this article, the authors have discussed in silico design novel griseofulvin derivatives with potential antibacterial activity based on molecular docking and molecular dynamics simulation study. The authors have made decent efforts to come up with a research article using in silico tools. In the reviewer’s opinion, the manuscript has been organized appropriately. The reviewer feels that a potential reader can easily understand the given details by consulting the main data outlined in the current version of the manuscript with few exceptions. The write-up is clear and to the point which is crucial for an article with few exceptions. Although there are several published articles closely related to this piece of study in general, the current version of the manuscript can bring a novelty in the existing literature however there are some laps that should be addressed before publication of this study. There are some suggestions related to the manuscript and authors should take into ‎account these suggestions before resubmission.

  1. In its current state, the title itself should be rephrased and needs careful attention.

The title is revised to “Computational Design of Novel Griseofulvin Derivatives Demonstrating Potential Antibacterial Activity: Insights from Molecular Docking and Molecular Dynamics Simulation”.

  1. The author revealed that their findings collectively suggest the promising potential of griseofulvin and its designed derivatives as effective antibacterial agents, particularly concerning their interaction with the ftsZ protein. Similarly, Table 2 shows the prediction of antibacterial and phosphatase inhibitor activities (Pa) for griseofulvin and derivatives G1-G6. The findings of this study have not been biologically validated not even a single type of testing ws performed. I would gladly reconsider revision if the author can provide biological validation of the claimed findings.

Although this special issue "Advances in Molecular Modeling, Docking and Simulations of Protein Structure" emphasizes the in silico analysis, the reviewer's suggestion on experimental evidence is highly relevant and well appreciated. In our next project, we plan to perform experimental studies on the designed griseofulvin derivatives based on the current findings.

  1. The introduction section lacks objectivity and should be revised with more clarity. While it is undeniable that theoretical studies play a critical role in accelerating drug development and conserving resources, the section could benefit from providing specific examples of such studies. For instance, it could mention instances where extensive theoretical studies have led to the qualification of biological molecules from hit to lead. This would add more weight to the argument and make the section more objective.

More details are added to the lines 88-95 to provide clarity regarding the purpose of this study.

  1. The authors hypothesized the notion that griseofulvin and its newly devised analogs could potentially interact favorably with the ftsZ protein in bacterial cells, given their structural resemblance to microtubules. Nevertheless, no empirical evidence/ biological testing was presented to verify this hypothesis in the current study.

The reviewer's suggestion on experimental evidence is highly relevant and well appreciated. As we mentioned above, we plan to perform experimental studies on the designed griseofulvin derivatives based on the current findings in the next project.

  1. In the opinion of the reviewer, a limited number of theoretical analogs were designed by the author, six in total. This low number is insufficient to generate a meaningful SAR for the generalization of activity. The authors should discuss the ‎limitations of being limited to such small theoretical analogs.‎

This limitation is mentioned and discussed in lines 362-371.

  1. A few types of amines were chosen for the design of G3-G6. What was the crucial factor in selecting these amines??

As we mentioned in lines 78-89, benzenesulfonyl hydrazones are known for their potential antibacterial activity, belonging to a class of compounds with diverse biological activities in medicine, primarily in the areas of antibacterial, antifungal, anticancer, and antidepressant effects. Aromatic amine 3,4-dichloroaniline is exclusively used as an intermediate in the chemical industry for synthesizing the antibacterial agent triclocarban, particularly effective against Gram-positive bacteria, such as Staphylococcus aureus. Reports suggest that the presence of a methylenedioxy ring in derivatives of methylenedioxy aniline may enhance their antibacterial activity. Notably, morpholine, a strong alkali, is a crucial precursor in antibiotic production. Taking into account the antibacterial potential of these compounds, we aimed to design novel griseofulvin analogs as antibacterial agents.

Reviewer 4 Report

Comments and Suggestions for Authors

This study helps improving the potential antibacterial activity of griseofulvin through a hybridization approach. New promising derivatives have been designed that target bacterial proteins.

Some minor editing is required for the presentation:

- First word "In" in the abstract should not be in bold.

- Some hyphen should be deleted in lines 48 & 50.

- Table 1 could be further explained, and presentation improved. Also, I'm guessing that Lipinski analysis is not used as a filter to disregard molecules but merely for illustrative properties. Maybe that could be highlighted. Moreover, have other LADME/toxicity filters/rules been considered?

- Tables 3 & 4 could be edited so the word "ciprofloxacin" fits in one line

- There is no "Discussion" section, although the results are discussed in the "Results" section. Given the number of steps, the lay-out of the manuscript seems more intuitive in the present form, but it does not follow the requirements.

Congratulations for your work. One can tell that lots of effort has been put into it.

Author Response

This study helps improving the potential antibacterial activity of griseofulvin through a hybridization approach. New promising derivatives have been designed that target bacterial proteins.

Some minor editing is required for the presentation:

- First word "In" in the abstract should not be in bold.

It has been fixed.

- Some hyphen should be deleted in lines 48 & 50.

They are deleted.

- Table 1 could be further explained, and presentation improved. Also, I'm guessing that Lipinski analysis is not used as a filter to disregard molecules but merely for illustrative properties. Maybe that could be highlighted. Moreover, have other LADME/toxicity filters/rules been considered?

More explanations are added to the lines 176-178.

- Tables 3 & 4 could be edited so the word "ciprofloxacin" fits in one line.

Thank you for your attention to the details. However, in the version that we submitted “ciprofloxacin” is in one line in both tables 3 and 4.

- There is no "Discussion" section, although the results are discussed in the "Results" section. Given the number of steps, the lay-out of the manuscript seems more intuitive in the present form, but it does not follow the requirements.

As you mentioned correctly, the results and discussions are combined to enhance clarity and facilitate understanding for readers.

Congratulations for your work. One can tell that lots of effort has been put into it.

Thank you so much.

Round 2

Reviewer 3 Report

Comments and Suggestions for Authors

The author has made some or partial modifications but excluded certain parts of the recommended experimentation that seem reasonable given the current situation. In Section 3, it would be better to use "Results and Discussion" instead of just "Results".

Author Response

The author has made some or partial modifications but excluded certain parts of the recommended experimentation that seem reasonable given the current situation. In Section 3, it would be better to use "Results and Discussion" instead of just "Results".

Thank you so much for your great comment. Section 3 is changed to "Results and Discussion" instead of "Results".